# The Influence of the Microbiome on Urological Malignancies: A Systematic Review

**DOI:** 10.3390/cancers15204984

**Published:** 2023-10-14

**Authors:** Joao G. Porto, Maria Camila Suarez Arbelaez, Brandon Pena, Archan Khandekar, Ankur Malpani, Bruno Nahar, Sanoj Punnen, Chad R. Ritch, Mark L. Gonzalgo, Dipen J. Parekh, Robert Marcovich, Hemendra N. Shah

**Affiliations:** 1Desai Sethi Urology Institute, University of Miami, Miller School of Medicine, Miami, FL 33136, USA; 2Stony Brook University Hospital, Stony Brook, NY 11794, USA

**Keywords:** microbiome, prostate cancer, bladder cancer, kidney cancer, testicular cancer, penile cancer

## Abstract

**Simple Summary:**

The human microbiome has become an increasingly important area of study in recent years, with growing evidence suggesting that it plays a critical role in numerous diseases, including those affecting the urinary tract. However, the specific role of the microbiome in malignant urologic diseases remains largely unknown. Our study aimed to synthesize current evidence regarding the relationship between the microbiome and these urologic conditions, such as prostate, bladder, kidney, penile, and testicular cancer. Our systematic review of 37 studies provides an up-to-date overview of the microbiome’s role in urologic health concerns. Using the present study as a guide, future studies have potential for microbiome-focused interventions to offer hope for presently unexplained medical conditions.

**Abstract:**

The microbiome, once considered peripheral, is emerging as a relevant player in the intricate web of factors contributing to cancer development and progression. These often overlooked microorganisms, in the context of urological malignancies, have been investigated primarily focusing on the gut microbiome, while exploration of urogenital microorganisms remains limited. Considering this, our systematic review delves into the complex role of these understudied actors in various neoplastic conditions, including prostate, bladder, kidney, penile, and testicular cancers. Our analysis found a total of 37 studies (prostate cancer 12, bladder cancer 20, kidney cancer 4, penile/testicular cancer 1), revealing distinct associations specific to each condition and hinting at potential therapeutic avenues and future biomarker discoveries. It becomes evident that further research is imperative to unravel the complexities of this domain and provide a more comprehensive understanding.

## 1. Introduction

Defining the term “microbiome” is complex due to the intricate relationships within microbial communities and their host organisms. Over time, numerous definitions have emerged, with some focusing on the community of microorganisms, others emphasizing their collective genomes, or others yet looking at the broader ecological context, including biotic and abiotic conditions. Despite the varying definitions, it is essential to view the microbiome as a dynamic ecosystem, highlighting the interconnections among microorganisms and their environment [1,2,3]. Extensive exploration of the role of the microbiome in cancer evolution has demonstrated that while the microbiome can affect cancer cells themselves, it can also modulate cancer immunosurveillance [4,5].

Cancer has an immense impact on international health and lifespan. An extensive analysis of cancer prevalence and mortality data from GLOBOCAN, encompassing 185 nations and 36 distinct types of cancer, estimated that in 2020 there were approximately 19.3 million cases of cancer and nearly 10 million deaths worldwide due to the disease. Alarming projections indicate that the worldwide toll of cancer is anticipated to rise to 28.4 million incidences by the year 2040 [6].

Worldwide in 2018, approximately 2.2 million cancer diagnoses were linked to infections. Moreover, it is believed that nearly one-fifth of all human cancers could possibly result from infectious pathogens [7].

The human microbiome, an expansive network of often overlooked organisms, plays a crucial role in our health and wellbeing. Individual differences in these microbiomes result in a distinctive, ever-changing microbial fingerprint [8]. These microbes have demonstrated various functions, including metabolism, local and systemic inflammation, and immunity. Many studies have pinpointed discrepancies in microbial fingerprints between healthy individuals and patients with urological malignancies. Such microbiomes have been found to host microbes that might contribute to the development of disease and provide insights about a person’s health and response to specific drugs, such as immune checkpoint inhibitors (ICIs) and tyrosine kinase inhibitors (TKIs).

## 2. Materials and Methods

On 28 March 2023, we conducted a comprehensive literature search on Ovid MEDLINE using the following keywords: “microbiota” AND “prostate cancer”, “microbiota” AND “bladder cancer”, “microbiota” AND “renal cancer”, “microbiota” AND “testicular cancer”, and “microbiota” AND “penis cancer”. The aim of the search was to identify relevant clinical microbiome studies associated with noninfectious malignant urological conditions that were published between 2013 and 2023. This systematic review was not registered.

The inclusion criteria for the purposes of this review included the following: all studies had to (i) be solely focused on assessing the relationships between the urine microbiome and non-infectious neoplastic urologic diseases; (ii) use human clinical samples; (iii) perform 16S ribosomal RNA (rRNA) gene sequencing through polymerase chain reaction (PCR) or next generation sequencing (NGS); (iv) be written in English.

This systematic review strictly adhered to the preferred reporting items for systematic reviews and meta-analyses (PRISMA) framework (Figure 1). Initially, after removing duplicate entries, a thorough evaluation of search results was carried out independently by three reviewers (BP, JGP, and MCS), who screened titles and abstracts for relevance. Following this, the chosen set of studies underwent a second screening process based on the full-text articles. Exclusions were made for review articles, editorials, comments, systematic reviews, or conference abstracts lacking complete data. Additionally, studies involving the gut microbiome, animal subjects, or in vitro experiments were also excluded. Since this study only utilized publicly available unidentified data from known publications, institutional review board approval or patient consent were not required.

In case of any disagreements during the review process, a fourth member (HNS) was consulted, and a consensus was diligently reached. In total, our analysis identified 37 studies that met the specified criteria and aligned seamlessly with the focus of our research [9,10,11,12,13,14,15,16,17,18,19,20,21,22,23,24,25,26,27,28,29,30,31,32,33,34,35,36,37,38,39,40,41,42,43,44,45]. These studies were meticulously categorized into four distinct groups: prostate cancer (12), bladder cancer (20), kidney cancer (4), and penis/testicular cancer (1), ensuring a structured approach to our findings (Table 1).

## 3. Discussion

### 3.1. Prostate Cancer

Prostate cancer (PCa), resulting in over 250,000 deaths worldwide annually, represents the prevailing form of nonskin malignancy and stands as the 6th foremost contributor to cancer-related fatalities in men, thereby engendering significant apprehension within the realm of public health [15,16]. Recent insights have increasingly implicated chronic inflammation as a crucial factor in the pathogenesis of PCa [46]. Given the well-established role of microorganisms in provoking prostatic inflammation, the question of whether a correlation exists between the microbiome and PCa has emerged [9,46].

The gastrointestinal microbiome stands as one of the most extensively investigated human microbiomes. Numerous studies have delved into its potential correlation with PCa pathogenesis, treatment response, and prognosis [47,48,49]. However, recent attention has shifted towards other microbiotas, such as the urinary microbiome, as the prostate–urine loop has exhibited promising outcomes. For instance, Alanee et al. reported that individuals diagnosed with PCa exhibited similar bacterial communities that clustered separately from those without cancer [14]. Additionally, it was observed that patients with PCa exhibited a significantly lower urinary alpha diversity in comparison to individuals without malignancy. Notably, patients with Gleason 7 cancer displayed a distinct urinary microbiome profile when contrasted with men who did not harbor malignancy. Conversely, fecal samples did not demonstrate any clustering or significant disparity in alpha diversity or Gleason staging concerning benign or malignant prostate biopsies [14].

Similarly, in a case-control study comparing men with benign and malignant prostate biopsies, noteworthy findings emerged. The study revealed that biopsies containing a higher degree of cancer exhibited a distinct urinary cluster of bacteria, including *Streptococcus anginosus*, *Anaerococcus lactolyticus*, *Anaerococcus obesiensis*, *Actinobaculum schaalii*, *Varibaculum cambriense*, and *Propionimicrobium lymphophilum* [13]. Nonetheless, no significant differences were found in the urinary alpha or beta diversity between the cases and controls. Furthermore, there was no disparity detected in the relative abundance of *Propionibacterium acnes*, a species previously associated with PCa [9,13].

Interestingly, a study that evaluated urine samples after digital rectal exam and prostatic secretions after prostatectomy found five genera of strict anaerobes (*Fenollaria*, *Peptoniphilus*, *Anaerococcus*, *Porphyromonas*, and *Fusobacterium*) that could serve as urinary biomarkers due to their association with PCa aggressiveness and biochemical recurrence [15], highlighting the great potential that the urinary microbiome could play in PCa diagnosis, staging, and even prognosis.

While bacteria are the most frequently detected germ in the human microbiome, investigations in the context of PCa have unveiled the presence of viruses, fungi, and even protozoa. For example, Wang et al. discovered that in the plasma microbiome of men with PCa, the beta diversity of the circulating fungal microbiome exhibited notable distinctions when compared to age- and race-matched healthy controls [50]. Moreover, their findings highlighted a significant enrichment of class *Sordariomycetes* among PCa patients with advanced pathological grades (pT3 or pT4) [50]. Banerjee et al. also reported a distinguished fungal microbiome when comparing the prostate specimens of patients with benign prostatic hyperplasia (BPH) and PCa, finding that the majority of the fungal signatures in PCa arose from the division *Ascomycota* (61%) [20]. Remarkably, the researchers also observed distinct viral compositions among the groups under investigation. However, what proved to be particularly intriguing was their ability to identify the integration of viral and microbial DNA into human chromosomes [20]. The study revealed a significant number of genomic integration sites, with human papillomavirus (HPV) 18 exhibiting the highest count, followed by Kaposi sarcoma-associated herpesvirus (KSHV) and HPV2. It is worth noting that these viruses have been previously linked to tumorigenesis [20].

A study conducted with a similar objective, comparing the microbiome in biopsies from men diagnosed with BPH and PCa, revealed a high prevalence of tumorigenic viruses, such as HPV and hepatitis B virus (HBV), in the PCa specimens. Furthermore, a robust association was established between the presence of these viruses and the elevated abundance of two distinct bacterial species, *Cupriavidus taiwanensis* and *Methylobacterium organophilum*, within diseased prostate lesions [16]. The notable prevalence of these tumorigenic viruses raises the question of whether there is an association between sexually transmitted infections and the inflammatory environment that contributes to prostate carcinogenesis. Miyake et al. elegantly answered this question by screening surgical and biopsy specimens from 45 patients with PCa (cases) and 33 patients with BPH (controls) against a panel of sexually transmitted infection-related organisms [51]. They found that among the tested organisms, *Mycoplasma genitalium* was significantly more prevalent in the PCa group and was associated with the extensiveness of the diseases. Nonetheless, it is important to note that no significant association was found between *Mycoplasma genitalium* and the grade of intraprostatic inflammation [51].

Another intriguing aspect potentially linking PCa and the microbiome pertains to whether the observed variations in incidence across races and geographical locations correspond to disparities in the microbiome’s composition. A study comparing fresh prostate biopsies from age-matched Africans and Australians reported that the overall bacterial composition was similar between groups, with the most abundant genera being *Escherichia*, *Propionibacterium*, and *Pseudomonas* [17]. A significantly higher bacterial richness was observed in the African group, suggesting that bacterially driven oncogenic transformation within the prostate microenvironment may contribute to the aggressive disease presentation commonly observed in Africans [17].

Interestingly, a parallel study conducted in a Chinese population revealed comparable findings, with *Escherichia*, *Propionibacterium*, and *Pseudomonas* emerging as the most abundant genera within the prostate [18]. Furthermore, they reported no discernible differences in the alpha diversity or relative abundances between the tumor and adjacent benign prostate tissue. Consequently, no definitive association could be established between the microbiome and the Gleason stage or the prostate-specific antigen (PSA) level [18]. Likewise, a study examining prostate cancer tumor, peri-tumor, and non-tumor tissues in white, nondiabetic, nonobese PCa patients yielded analogous results [11]. These findings underscore the need for further investigation to elucidate the intricate interplay between the microbiome, prostate cancer, race, and geographical location. Additional studies are necessary to consolidate and strengthen the existing body of evidence.

Prostate cancer showcases a distinctive signature within the urine, plasma, and prostate microbiotas. Further, PCa has demonstrated variations in terms of alpha diversity, beta diversity, and species composition when compared to other genitourinary malignancies, such as bladder and kidney cancer [26]. The unique microbiome alterations associated with it have also been implicated in metagenomic and metabolic changes that have even been linked with modifications in inflammatory responses [19]. It is important to acknowledge the inherent challenges in generalizing results due to the considerable inter- and intra-variability of the microbiome. However, the observed associations between PCa and the microbiome serve as compelling impetus to further advance research in this field. By doing so, we may eventually harness the potential of the microbiome as a diagnostic, prognostic, and therapeutic tool.

### 3.2. Bladder Cancer

In 2020, there were a total of 573,278 new urothelial bladder carcinoma cases and 212,536 deaths, positioning it as the 10th most common form of cancer diagnosed globally [52]. The bladder was previously thought to be a sterile environment, but emerging research has revealed the presence of a distinct microbial community, known as the bladder microbiome [53].

Several studies have reported dysbiotic changes in the bladder microbiome of individuals with bladder cancer. These changes are characterized by a decrease in microbial diversity and alterations in the abundance of specific bacterial taxa. It was found that patients with bladder cancer had lower alpha diversity than non-cancer patient controls [22,23,32]. Ma et al. stated that smoking is one of the main factors affecting the urinary tract’s microbial composition on a beta diversity analysis. The top five species in smokers were *Bacteroidetes*, *Bacteroides*, *Bacteroidales*, *Clostridia*, and *Clostridiales* [29]. In addition to the identification of specific biomarkers in the bladder, the microbiome may have diagnostic and prognostic implications for bladder cancer. Studies have shown the potential of microbial signatures as non-invasive biomarkers for early detection and monitoring of bladder cancer. It is well known that infections caused by *Schistosoma haematobium*, a species of parasitic blood fluke, are associated with the development of squamous cell carcinoma of the bladder due to chronic inflammation [54]. Schistosomiasis progression has also been associated with other organisms, like *Fusobacterium*, *Sphingobacterium*, *Bacterioides*, and *Enterococcus*, known as mediators of inflammatory and immunologic reactions [55].

Although the genitourinary microbiome has a possible role in bladder cancer progression, it also has a role in treatment. The Bacillus Calmette–Guerin (BCG) vaccine, known to prevent disseminated tuberculosis, has been used to treat both intermediate and high-grade non-muscle invasive bladder cancer (NMIBC), as well as specific cases of low-grade NMIBC (large, multifocal, recurrent) [53,56]. This vaccine is identified as a *Mycobacterium*, an *Actinomycete* that both activates the immune system and reacts with tumor cells, causing apoptosis, necrosis, and oxidative stress [57]. A study showed a significantly higher abundance of Actinobacteria in non-neoplastic bladder mucosa samples than in tumor tissues, supporting the hypothesis that microbiota rich in *Actinomycetes* are related to the lower incidence of bladder cancer in women and, therefore, may have a preventive effect against this type of cancer [24].

Comparative analysis of the bladder microbiome between cancer patients and controls has also provided important insight into how diverse the bacterial species are and how they are related to the progression of cancer. In a study completed in the Czech Republic, which collected and compared urine samples of 34 men with bladder cancer and 29 control patients and described bladder cancer patients at the phylum level, Firmicutes, Proteobacteria, Actinobacteria, and Bacteroidetes were the most predominant [22]. Mansour et al. described a median abundance of *Cyanobacteria*, *Staphylococcus*, and *Corynebacterium* as well as the previously mentioned microbial groups. Similar findings were noted in two studies from Spain and United States of America: Parra-Grande et al. and Zeng et al., respectively [24,34]. Likewise, some studies reported a high abundance of the phyla Bacteroides, Faecalibacterium, Bacilliales, Clostridium, and Actinobacteria and at the genus level, *Streptococcus*, *Lactobacillus*, and *Prevotella* [23,36,37,39]. Interestingly, there is an association between the increased abundance of Fusobacterium and reduced levels of Lactobacillus and bladder cancer [54]. Bučević et al. state that *Fusobacterium nucleatum* may be a pro-tumorigenic pathogen based on their findings in 11 out of 42 patients [39].

Compositionally, Perdezoli et al. found that the urinary microbiome was dominated by the members of the three major bacterial phyla: Proteobacteria, Firmicutes, and Bacteroidetes, which accounted for 80% in males and 90% in females of all identified taxa [30]. Notably, they highlighted the presence of the order *Opitutales* and its subordinate family, *Opitutaceae*, as well as the isolated class *Acidobacteria* in males. In females, they identified the genus *Klebsiella*, belonging to the family *Enterobacteriaceae*. Furthermore, their research revealed a significantly higher abundance of the genus *Burkholderia* in both male and female tissues, strongly suggesting its association with bladder cancer [30].

The microbiome’s composition presented significant differences according to the tumor tissue and its grade. Comparing muscle invasive bladder cancer (MIBC) and NMIBC, both groups presented Firmicutes, Proteobacteria, Actinobacteria, and Fusobacterium as the main phylum level [39,40]. Hussein et al. studied 42 patients with bladder cancer (MBIC 14 and NMIBC 29), finding that *Haemophilus* and *Veillonella* were significantly more abundant in the urine of the MIBC patients, while *Cupriavidus* was present in the NMIBC patients; when subclassified by sex, they found *Actinobacteria* predominantly in men and *Bacteriodetes* in females [31]. Another study with 899 patients from China found that *Eubacterium CAG-581*, *Bacteroides sp 43-47FAA*, and *Flavobacteriales* were predominant in the NMIBC group, reflecting a strong association with a decreased survival rate. Comparing NMIBC patients who responded to BCG with those who did not, *Serratia* and *Brochothrix*, *Negativicoccus*, *Escherichia-Shigella*, and *Pseudomonas* were significantly more abundant in the respondent group [21,31].

It has also been suggested that the microbiota may upregulate programmed death-ligand 1 (PD-L1) expression in epithelial tumors, promoting an immune escape [58]. Chen et al. stated that the rates of recurrence and progression of NMIBC are higher in PD-L1 positive patients rather than in PD-L1 negative patients. They found that PD-L1 positive patients with NMIBC had abundant *Leptotrichia* which is closely related to *Fusobacterium*, previously described as a pro-tumorigenic pathogen. On the other hand, *Prevotella* was abundant in PD-L1 negative patients with NMIBC [25].

Further research is needed to elucidate the complex relationship between the bladder microbiome and bladder cancer. Longitudinal studies, functional characterization of microbial communities, and investigation of host–microbe interactions will contribute to a deeper understanding of the role of the bladder microbiome in disease development and treatment.

### 3.3. Kidney Cancer

Kidney cancer is ranked as the 14th most prevalent neoplasm on a global scale, with an incidence rate of 4.5 cases per 100,000 individuals [6,59]. Renal cell carcinoma (RCC) most commonly encompasses the primary subtypes of clear cell (ccRCC), papillary, and chromophobe [60]. Generally, risk factors associated with this ailment include smoking, obesity, and hypertension [61]. While research on the impact of microbiomes on this condition remains limited, most studies focus on the gut microbiota [62,63,64,65,66,67,68,69,70,71]. It is widely recognized that gastrointestinal conditions influence the kidney, and investigating the gut microbiome offers advantages due to the relatively accessible nature of this microbiome as compared to the kidney itself, facilitating sample collection and analysis. However, emerging studies on the kidney microbiome itself hold promise for providing further insights into kidney cancer and its associated conditions.

Three studies investigated retrospectively the role of the microbiome in RCC by examining mainly neoplastic kidney tissue and adjacent normal tissue (ANT). These studies utilized 16S rRNA sequencing as a method to extract bacterial DNA. In a pilot study conducted by Heidler et al., five patients with renal cancer (including four with RCC and one with chromophobe cell carcinoma) who underwent radical nephrectomy were assessed [43]. There was a significant difference in microorganisms between the malignant and benign tissue, with the malignant tissue exhibiting more diversity.

Another study by Wang et al., involving 24 RCC patients, also revealed a distinct composition of the bacterial community in RCC and normal tissues [42]. However, the authors found a decrease in species diversity within the RCC tissue. Both studies reached a consensus that Firmicutes, Proteobacteria, Bacteroidetes, and Actinobacteria were the predominant phyla in both cancerous and non-cancerous renal tissues. Additionally, both studies identified the bacterial family *Comamonadaceae* as significantly more abundant in the control group [42,43].

Furthermore, Wang et al. highlighted the accuracy of prediction of RCC of certain microbiomes, where high accuracy was found for the class *Chloroplast* and the order *Streptophyta*, while moderate accuracy for *Deinococcus* and *Phyllobacterium* [42]. These findings indicate the potential for utilizing the microbiome as a reliable biomarker in various areas, such as treatment response monitoring, early detection, and prognostic assessment. Additionally, an assessment of tumor grade in neoplasia revealed an interesting trend [42]. It was observed that there is a decreasing trend in microbiota richness as the tumor grade increases. Moreover, it was found that *Rhodoplanes* were significantly more abundant in grade I RCC samples as compared to grade II samples. These findings provide valuable insights into the potential utilization of the microbiome as a risk stratification tool.

Microbiome-mediated modulation may impact diverse cellular processes and metabolic activities, potentially influencing cancer development and progression. Wang et al. investigated this relationship between the microbiome and RCC to comprehend the immune response within the tumor microenvironment [42]. In malignant tissues, they identified alterations in pathways related to membrane transport, transcription, and cell growth and death, while in ANT tissues, the affected pathways were cell motility, signal transduction, metabolism, metabolism of cofactors and vitamins, energy metabolism, and the endocrine system. Further investigations into the specific mechanisms underlying these alterations could offer promising insights into novel therapeutic approaches targeting the microbiome to modulate immune responses and improve patient outcomes in kidney cancer.

Given the significant roles that various receptors play in the development, progression, and targeted treatment of RCC, it is imperative to comprehend the association between the microbiome and these pertinent receptors. Liss et al. conducted a study to investigate the interaction between the microbiome and PD-L1 receptor expression in relation to venous tumor thrombus (TT) in six patients with ccRCC [41]. The study involved analyzing three samples: tumor, ANT, and TT. The findings revealed a greater diversity of microbiota in the renal tumor tissue as compared to ANT and TT. Specifically, they identified *Micrococcus luteus*, *Fusobacterium nucleatum*, *Streptococcus agalactieae*, and *Corynebacterium diphtheriae* in the tumor microbiome. Moreover, the authors discovered a correlation between the presence of aggregated oral microbiome within primary renal cancer tumors and its potential interaction with the tumor microenvironment. Patients with oral microbiome aggregation within the tumor exhibited higher expression of the PD-L1 receptor within the TT. On the other hand, in a study conducted by Najm et al., the expression of succinate receptor 1 (SUCNR1) and its impact on ccRCC and papillary RCC were investigated using the cBioPortal platform (http://cbioportal.org) [44,72,73,74,75]. The findings revealed that the SUCNR1 receptor is associated with distinct microbiome signatures in each RCC subtype, leading to different disease-specific survival predictions. Specifically, in ccRCC, high SUCNR1 receptor expression along with the presence of *Candidatus nitrosopelagicus* and *Indibacter bacteria* correlated with better survival outcomes. Conversely, the presence of *Anoxybacillus* and *Selenomonas* bacteria, along with low SUCNR1 receptor expression, was associated with poor disease-specific survival in ccRCC. In papillary RCC, high SUCNR1 receptor expression correlated with the presence of *Apibacter* bacteria, which was linked to a worse prognosis. Notably, the study highlights the importance of understanding these microbiome profiles and their relationship with SUCNR1 receptor expression in guiding future targeted therapies for improved patient selection and outcomes.

### 3.4. Penile Cancer

When it comes to the male genital organ and the microbiome context, the existing literature focuses on circumcision and/or sexually transmitted diseases [76,77,78]. There is a notable absence of studies exploring the connection between the penile microbiome and penile cancer. Investigating the penile microbiome and its potential impact on penile cancer could provide valuable insights and benefits.

### 3.5. Testicular Cancer

Testicular cancer, although rare with a global incidence rate of 0.4%, is the most common malignancy in young men [6,79]. Previous studies have explored the microbiome of seminal plasma and testicular tissue in benign conditions, revealing the presence of *Actinobacteria*, *Bacteroidetes*, *Firmicutes*, and *Proteobacteria* as the most commonly found phyla [80,81]. However, only one study has specifically investigated microbiomes in the context of testicular cancer. Mørup et al. used small RNA sequencing to analyze seminal plasma samples from patients with testicular germ cell tumors (TGCT) and germ cell neoplasia in situ (GCNIS) [45]. Overall, the most prevalent microbes in the study were *Alteromonas mediterranea*, *Falconid herpesvirus* 1, and *Stigmatella aurantiaca*. When assessing the association with TGCT/GCNIS or GCNIS only, the authors identified *Acaryochloris marina*, *Halovirus HGTV-1*, *Thermaerobacter marianensis*, *Thioalkalivibrio sp. K90mix*, *Burkholderia sp. YI23*, and *Desulfurivibrio alkaliphilus* as the most common microorganisms. Conversely, *Streptomyces phage VWB* was significantly associated with subjects without malignancy. Moreover, after a comparison between seminoma and non-seminoma patients, no significant differences were observed in terms of microbiome composition.

The use of seminal plasma in testicular cancer research offers several advantages. It is a non-invasive and accessible sampling method, often collected as part of routine fertility testing or semen analysis. By analyzing the seminal plasma microbiome, valuable insights into the local microenvironment of the male reproductive system are provided. This approach holds potential for diagnostic value, as it may help identify biomarkers or specific microbial signatures associated with the disease. Such findings could contribute to the development of non-invasive screening methods and improved diagnostic tools. However, further studies are necessary to compare the microbiomes of malignant and benign testicular tissue and to investigate the relationship between the testicular and seminal plasma microbiomes. Additionally, exploring the microbiome of seminal vesicles could provide additional insights into the microbiome dynamics in this field.

## 4. Future Directions

Scientific understanding within the sphere of the urogenital microbiome and its association with malignant urological conditions is currently in an evolving phase, with limited concrete clinical insights to date. While the forefront of microbiome research has predominantly centered on the gut microbiome, it is reasonable to anticipate that similar valuable discoveries will emerge within the urogenital microbiome domain. Advancements in methodologies for assessing and gathering urogenital microbiome data are imperative to unlock this potential and pave the way for future insights. There is potential for microbiome-focused interventions to offer hope for presently unexplained medical conditions. Conducting long-term studies to track changes in the urinary microbiome over time can offer valuable insights into the development and resolution of diseases, leading to substantial advancements in clinical practice.

Furthermore, while many studies commonly rely on the valuable 16S rRNA technology to gain insights into the general composition of the microbiome, it often falls short in offering an in-depth characterization of specific species and strains. In this regard, future research endeavors employing total RNA/DNA sequencing hold the promise of delivering more comprehensive insights into the composition of the microbiome and its relationship with urological malignancies.

While our study primarily focused on analyzing the microbiome profiles of specific urologic tumors individually, it becomes increasingly pertinent to identify commonalities and disparities across different tumor types. Future studies should prioritize a comparative approach that systematically investigates the microbiome signatures shared among these urologic tumors and the unique microbial fingerprints associated with each. Such investigations hold the potential to unveil overarching microbiome patterns that may transcend tumor boundaries while shedding light on specific microbiome nuances that contribute to tumor heterogeneity. Furthermore, by elucidating the common and differential microbial factors at play, we may uncover opportunities for cross-disciplinary collaboration and the development of innovative diagnostic and therapeutic strategies that benefit patients across a spectrum of urological cancers.

## 5. Conclusions

The microbiome in urological oncology offers a glimpse into a future where personalized medicine harnesses the power of microbiota as diagnostic tools, prognostic indicators, and even therapeutic targets. However, presently we are only scratching the surface, and future investigations are essential to unlock the full potential of the urogenital microbiome in transforming the landscape of urological oncology. With continued research and advancements in methodologies, we stand poised to uncover new insights, enhance patient care, and ultimately improve outcomes for those affected by malignancies.

## Figures and Tables

**Figure 1 cancers-15-04984-f001:**
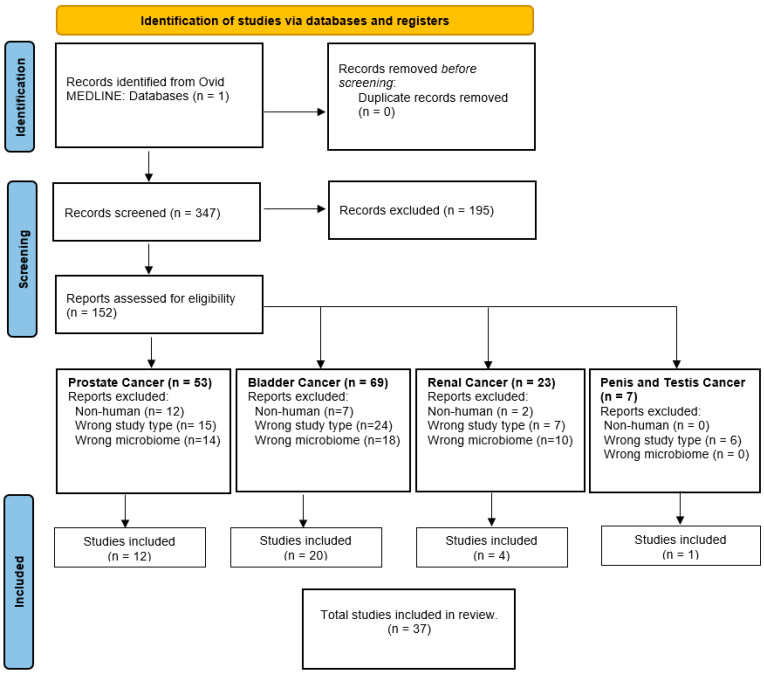
PRISMA 2020 flow diagram for systematic reviews.

**Table 1 cancers-15-04984-t001:** Prostate, bladder, kidney, and testicular cancer studies with related microbiome features.

Study	Number of Participants	Specimen	Sequencing Technique
**Prostate cancer**			
Cohen et al. (2005) [9]	34 PCa	Tumor tissue	16S rRNA seq
Yu et al. (2015) [10]	13 PCa and 21 BPH	Urine sampling, prostatic secretions, and seminal fluid	PCR-DGGE
Cavarretta et al. (2017) [11]	6 PCa	Tumor tissue, peri-tumor, and ANT	NGS
Yow et al. (2017) [12]	10 PCa	Tumor tissue	16S rRNA seq
Shrestha et al. (2018) [13]	65 benign tissue, 65 PCa	Urine sampling	16S rRNA/16S rDNA
Alanee et al. (2019) [14]	30 PCa	Urine and fecal sampling	NGS
Hurst et al. (2022) [15]	300 PCa	Urine sampling, prostate secretions, and tumor tissue	16S rRNA seq
Sarkar et al. (2022) [16]	16 PCa and 15 BPH	Tumor tissue	16S rRNA seq
Feng et al. (2019) [17]	28 PCa	Tumor tissue	NGS
Feng et al. (2019) [18]	65 PCa	Tumor tissue and ANT	DNA and RNA seq
Salachan et al. (2022) [19]	94 PCa	Tumor tissue	RNA seq
Banerjee et al. (2019) [20]	50 PCa and 15 benign tissues	Tumor tissue and benign tissue	NGS
**Bladder cancer**			
Zhang et al. (2023) [21]	51 NMIBC and 47 controls	Mid-stream urine and tumor tissue	16S rRNA seq
Hrbacek et al. (2023) [22]	34 NMIBC and 29 controls	Mid-stream urine	16S rRNA seq
Mansour et al. (2022) [23]	55 bladder cancer and 12 controls	Urine sampling and tumor tissue	16S rRNA seq
Parra-Grande et al. (2022) [24]	32 bladder cancer and 26 controls	Tumor tissue and ANT	16s rRNA seq
Chen et al. (2022) [25]	28 NMIBC	Mid-stream urine and tumor tissue	16s rRNA seq
Ahn et al. (2022) [26]	30 bladder cancer	Urine sampling	16s rRNA seq
Qiu et al. (2022) [27]	40 NMIBC	Mid-stream urine and tumor tissue	16s rRNA seq
Nadler et al. (2021) [28]	10 NMIBC	Tumor tissue	16s rRNA seq
Ma et al. (2021) [29]	15 bladder cancer and 11 controls	Mid-stream urine	16s rRNA seq
Pederzoli et al. (2020) [30]	49 bladder cancer and 59 controls	Urine sampling and tumor tissue	16s rRNA seq
Hussein et al. (2021) [31]	43 bladder cancer and 10 controls	Urine sampling	16s rRNA seq
Chipollini et al. (2020) [32]	27 bladder cancer and 10 controls	Mid-stream urine	16s rRNA seq
Hourigan et al. (2020) [33]	22 bladder cancer	Mid-stream urine	16s rRNA seq
Zeng et al. (2020) [34]	62 (51 NMIBC/11 MIBC) and 19 controls	Mid-stream urine	16s rRNA seq
Mansour et al. (2020) [35]	6 NMIBC and 4 MIBC	Urine sampling and tumor tissue	16s rRNA seq
Bi et al. (2019) [36]	29 bladder cancer and 26 controls	Mid-stream urine	16s rRNA seq
Liu et al. (2019) [37]	22 NMIBC/MIBC and 12 controls	Tumor tissue and ANT	16s rRNA seq
Moynihan et al. (2019) [38]	8 bladder cancer and 32 controls	Mid-stream urine	16s rRNA seq
Bučević et al. (2018) [39]	17 NMIBC and 19 controls	Mid-stream urine and tumor tissue	16s rRNA seq
Wu et al. (2018) [40]	31 (26 NMIBC/5 MIBC) and 18 controls	Mid-stream urine	16s rRNA seq
**Kidney cancer**			
Liss et al. (2020) [41]	6 ccRCC	ANT, tumor tissue, and thrombus	16S rRNA seq
Wang et al. (2021) [42]	24 RCC	Tumor tissue and ANT	16S rRNA seq
Heidler et al. (2020) [43]	5 RCC	Tumor tissue and benign renal tissue	16S DNA seq
Najm et al. (2022) [44]	20 RCC	-	log RNA Seq CPM
**Testicular cancer**			
Mørup et al. (2022) [45]	5 GCNIS-only, 18 TGCT, and 25 controls	Semen	RNA seq

ANT: adjacent normal tissue; BPH: benign prostatic hyperplasia; ccRCC: clear cell renal carcinoma; GCNIS: germ cell neoplasia in situ; MIBC: muscle invasive bladder cancer; NGS: next-generation sequencing; NMIBC: non-muscle invasive bladder cancer; PCa: prostate cancer; PCR-DGCE: polymerase chain reaction/denaturing gradient gel electrophoresis; RCC: renal carcinoma; rDNA: ribosomal DNA; rRNA: ribosomal RNA; Seq: sequencing; TGCT: testicular germ cell tumor.

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
