# Peer review of "The Influence of the Microbiome on Urological Malignancies: A Systematic Review"

_cancers, 2023, doi:10.3390/cancers15204984_

Round 1

Reviewer 1 Report

The manuscript presents published data on a new field, the role of microbiome in urological cancers. It is an innovative and important issue that must be widely disseminated to all stakeholders.

The review is organised well and offers significant insights on the issue.

There is only one major point that needs to be addressed in order to improve the quality of the manuscript.

The review article does not discuss the methods used to characterise the microbiome in clinical samples. This is a serious limitation in my opinion.
The vast majority of the cited articles use 16S rRNA sequencing which is
useful for obtaining a general composition of the microbiome but does
not allow in-depth characterisation of species and strains. The authors
should clearly state that in the future the use of RNAseq methods will
offer more significant insights on the microbiome composition and its
relation to urological malignancies.

The English language quality is good and requires minor editing.

Author Response

We thank the reviewer for the valuable comments. We agreed with the statement and included the following paragraph in the Future Directions Section of our manuscript.

Furthermore, while many studies commonly rely on the valuable 16S rRNA technology to gain insights into the general composition of the microbiome, it often falls short in offering an in-depth characterization of specific species and strains. In this regard, future research endeavors employing total RNA/DNA sequencing hold the promise of delivering more comprehensive insights into the composition of the microbiome and its relationship with urological malignancies.”

Reviewer 2 Report

I would like to commend the authors for their comprehensive and insightful review on the role of the microbiome in urological malignancies. The manuscript provides a valuable contribution to the field of cancer research, particularly in highlighting the often underestimated influence of microorganisms in cancer development and progression.

The introduction effectively sets the stage by emphasizing the growing significance of the microbiome in cancer, specifically in the context of urological malignancies. The focus on the gut microbiome and the limited exploration of urogenital microorganisms is a pertinent issue that warrants attention.

The systematic review's scope, which encompasses various urological cancers such as prostate, bladder, kidney, penile, and testicular cancers, demonstrates the authors' commitment to providing a comprehensive overview of the subject matter. The inclusion of 37 studies further strengthens the rigor and depth of the analysis.

The manuscript's findings, which reveal distinct associations specific to each urological cancer type, are intriguing. This highlights the potential for targeted therapeutic interventions and the discovery of novel biomarkers, which could significantly impact the field of oncology.

The conclusion appropriately emphasizes the necessity for further research in this area. It underscores the complexity of the microbiome's role in urological malignancies and encourages researchers to delve deeper into this domain to gain a more comprehensive understanding.

Author Response

Reply- We thank reviewers for their encouraging comments regarding our manuscript.

Reviewer 3 Report

The microbiome, once considered peripheral, is emerging as a relevant player in the intricate web of factors contributing to cancer development and progression. These often overlooked microorganisms, in the context of urological malignancies, have been investigated primarily focusing on the gut microbiome, while exploration of urogenital microorganisms remains limited. In this manuscript, the authors systematic review delves into the complex role of these understudied actors in various neoplastic conditions, including prostate, bladder, kidney, penile, and testicular cancers. Their analysis found a total of 37 studies, with prostate cancer-12, bladder cancer-20, kidney cancer-4, penile/testicular cancer-1, which reveals distinct associations specific to each condition, hinting at potential therapeutic avenues and future biomarker discoveries. This is an interesting topic. But I have several following concerns:

1. More articles with different definitions of "microbiome" should be cited.

2. The authors only analyzed the correlation between different urologic tumors and their own microbiome changes, but did not summarize the common and differences of microbiome between different tumors. The authors are advised to add these comparisions and discusssions.

3. Abbreviations should be defined When thery appears in the first time. Such as "PSA"

4. In Lines 217-218, 222-226, and other places. The category "phylum" does not need to be italicized.

5. Please define or explain "NMBIC" and "MIBC".

6. Figure 1 should be inserted in the text rather than in the supplementary material.

7. Please unify the format of references in the article, including the author's name, the case of words in the title of the article, the writing of the name of the journal, and the page number.

Minor editing of English language required.

Author Response

  1. More articles with different definitions of "microbiome" should be cited.

Reply – We thank the reviewer for the suggestion. We included new references vital to the comprehension and discussion of Microbiome.

  1. The authors only analyzed the correlation between different urologic tumors and their own microbiome changes but did not summarize the common and differences of microbiome between different tumors. The authors are advised to add these comparisons and discussions.

Reply - We thank the reviewer for the comment. We recognize the importance of expanding our analysis towards the evaluation of microbiome between different tumors. However, the knowledge associated in this field is still very limited. To address this gap in knowledge we included a paragraph for this topic in the future directions of our revised manuscript

"While our study primarily focused on analyzing the microbiome profiles of specific urologic tumors individually, it becomes increasingly pertinent to identify commonalities and disparities across different tumor types. Future research endeavors should prioritize a comparative approach that systematically investigates the microbiome signatures shared among these urologic tumors and the unique microbial fingerprints associated with each. Such investigations hold the potential to unveil overarching microbiome patterns that may transcend tumor boundaries while shedding light on specific microbiome nuances that contribute to tumor heterogeneity. Furthermore, by elucidating the common and differential microbial factors at play, we may uncover opportunities for cross-disciplinary collaboration and the development of innovative diagnostic and therapeutic strategies that benefit patients across a spectrum of urological cancers."

  1. Abbreviations should be defined when they appear in the first time. Such as "PSA"

Reply – We thank the reviewer for the comment. We included definition for all abbreviations in the manuscript.

  1. In Lines 217-218, 222-226, and other places. The category "phylum" does not need to be italicized.

Reply - We thank the reviewer for the comment. The modifications were included.

  1. Please define or explain "NMBIC" and "MIBC".

Reply – Thank you the reviewer for the comment. NMIBC stands for non-muscle invasive bladder cancer, while MICB refers to muscle invasive bladder cancer. We have included the definitions in the manuscript. NMBIC was a typing mistake that was corrected for NMIBC. 

  1. Figure 1 should be inserted in the text rather than in the supplementary material.

Reply – We thank the reviewer for the suggestion. We included the figure in the manuscript.

  1. Please unify the format of references in the article, including the author's name, the case of words in the title of the article, the writing of the name of the journal, and the page number.

Reply – We thank the reviewer for the comment. We modified the references according to journal guidelines.

Reviewer 4 Report

The authors present a review article on the relationship between the urogenital microbiome and urologic malignancies. The systematic review adheres to the PRISMA framework; the manuscript is well-written; and the results are interesting. This will serve as a useful roadmap for researchers in this space. The article might be improved by considering the following minor comments.

Methods: Paragraph 2: In the roman numeral count, the authors seem to have skipped “iv.”

There is no “Figure 1” in my PDF. I am not sure if that is a user error on my part, but the actual figure should be included in the final publication.

Discussion: Prostate cancer (3.1): Penultimate paragraph: The term “Caucasian” as a synonym for European has fallen out of favor. Its use has roots in disproven theories on biological race (https://www.nature.com/articles/d41586-021-02288-x). I recommend against using this term unless necessary.

Discussion: Bladder cancer (3.2): 3rd paragraph: The authors state that BCG is used for high-grade NMIBC. Current AUA guidelines recommend BCG for intermediate and high-risk NMIBC, which often includes low-grade NMIBC (large, multifocal, recurrent).

Discussion: Bladder cancer (3.2): Penultimate paragraph: Sentence 2: I think “NMIB” is a typo for “NMIBC.”

Discussion: Bladder cancer (3.2): 5th paragraph: Sentence 2: This sentence is hard to follow. I am not sure this is a sentence (where is the predicate?). Please consider rewording and/or dividing it into multiple sentences.

Author Response

Methods: Paragraph 2: In the roman numeral count, the authors seem to have skipped “iv.”

Reply – We thank the reviewer for bringing this point to our attention. The modification was included in the manuscript.

There is no “Figure 1” in my PDF. I am not sure if that is a user error on my part, but the actual figure should be included in the final publication.

Reply – In the revised manuscript, we included the figure in the text instead of supplementary.

Discussion: Prostate cancer (3.1): Penultimate paragraph: The term “Caucasian” as a synonym for European has fallen out of favor. Its use has roots in disproven theories on biological race (https://www.nature.com/articles/d41586-021-02288-x). I recommend against using this term unless necessary.

Reply – We thank the reviewer for the comment. We removed the term from our manuscript.

Discussion: Bladder cancer (3.2): 3rd paragraph: The authors state that BCG is used for high-grade NMIBC. Current AUA guidelines recommend BCG for intermediate and high-risk NMIBC, which often includes low-grade NMIBC (large, multifocal, recurrent).

Reply – We thank the reviewer for the comment. We modified the manuscript and added a reference of the guideline

“The Bacillus Calmette–Guerin (BCG) vaccine, known to prevent disseminated tuberculosis, has been used to treat both intermediate and high-grade non–muscle invasive bladder cancer (NMIBC), as well as specific cases of low-grade NMIBC (large, multi-focal, recurrent)”

Discussion: Bladder cancer (3.2): Penultimate paragraph: Sentence 2: I think “NMIB” is a typo for “NMIBC.”

Reply – We thank the reviewer for the comment. The modification was done.

Comments on the Quality of English Language

Discussion: Bladder cancer (3.2): 5th paragraph: Sentence 2: This sentence is hard to follow. I am not sure this is a sentence (where is the predicate?). Please consider rewording and/or dividing it into multiple sentences.

Reply – We agreed with the reviewer comment. We rewrote the sentence to improve the understanding of our findings.

Notably, they highlighted the presence of the order Opitutales and its subordinate family, Opitutaceae, as well as the isolated class Acidobacteria in males. In females, they identified the genus Klebsiella, belonging to the family Enterobacteriaceae. Furthermore, their research revealed a significantly higher abundance of the genus Burkholderia in both male and female tissues, strongly suggesting its association with bladder cancer

Round 2

Reviewer 1 Report

My comments are fully adressed.

Reviewer 3 Report

The authors have addressed all my comments. I recommend accepting this manuscript in current form.